# Extracellular Vesicle-Mediated Delivery of AntimiR-Conjugated Bio-Gold Nanoparticles for In Vivo Tumor Targeting

**DOI:** 10.3390/pharmaceutics17081015

**Published:** 2025-08-05

**Authors:** Parastoo Pourali, Eva Neuhöferová, Behrooz Yahyaei, Milan Svoboda, Adéla Buchnarová, Veronika Benson

**Affiliations:** 1Department of Chemistry, University of Wyoming, 1000 E. University Ave., Laramie, WY 82071, USA; ppourali@uwyo.edu; 2Institute of Microbiology, Czech Academy of Sciences, 142 20 Prague, Czech Republic; 3Department of Medical Sciences, Sha.C., Islamic Azad University, Shahrood 36199-43189, Iran; behroozyahyaei@yahoo.com; 4Department of Medical Sciences, Nanoparticle Research Center in Medicine, Sha.C., Islamic Azad University, Shahrood 36199-43189, Iran; 5Institute of Analytical Chemistry, Czech Academy of Sciences, 602 00 Brno, Czech Republic; svoboda@iach.cz; 6Faculty of Health Studies, Technical University of Liberec, 460 01 Liberec, Czech Republic; adela.buchnarova@tul.cz

**Keywords:** extracellular vesicles, biologically produced gold nanoparticles, RNA delivery, cancer tissue targeting, in vivo application

## Abstract

**Background/Objectives:** Extracellular vesicles (EVs) are involved in cell-to-cell communication and delivery of signaling molecules and represent an interesting approach in targeted therapy. This project focused on EV-mediated facilitation and cell-specific delivery of effector antimiR molecules carried by biologically produced gold nanoparticles (AuNPs). **Methods:** First, we loaded EVs derived from cancer cells 4T1 with AuNPs-antimiR. The AuNPs were also decorated with or without transferrin (Tf) molecules. We examined parental cell-specific delivery of the AuNPs-Tf-antimiR within monocultures as well as co-cultures in vitro. Subsequently, we used autologous EVs containing AuNPs-Tf-antimiR to target tumor cells in a xenograft tumor model in vivo. Efficacy of the antimir transfer was assessed by qPCR and apoptosis assessment. **Results:** In vitro, EVs loaded with AuNPs-antimiR were internalized only by the parental cells and the AuNPs-antimiR transfer was successful and effective only in EVs that were decorated with Tf. We achieved effective delivery of the antimiR molecule into cancer cells in vivo, which was proved by specific silencing of the target oncogenic miRNA as well as induction of cancer cells apoptosis. **Conclusions:** EVs represent an interesting and potent way for targeted cargo delivery and personalized medicine. On the other hand, there are various safety and efficacy challenges that remain to be addressed.

## 1. Introduction

The in vivo application of therapeutic nucleic acids (NAs) faces challenges, such as susceptibility to nuclease degradation, inefficient or non-specific cellular uptake, rapid clearance from the bloodstream, and low endosomal escape efficacy. Although there are various NAs delivery methods, sustainable and effective therapeutic carriers remain to be developed. For example, viral vectors, while effective, are associated with high production costs and unfavorable inflammatory responses. Nonviral alternatives, such as lipid nanoparticles, also present challenges, including activation of the immune system through complement activation [1], aggregation in the liver, and insufficient endosomal escape [2].

Another promising approach involves the use of extracellular vesicles (EVs) loaded with therapeutic NAs [3]. EVs naturally mediate intercellular communication, delivering signaling molecules without degradation or immune system recognition, owing to their surface proteins and specific cellular receptors [4]. It has been demonstrated that exosomes released by certain tumor cells contain tumor-specific antigen(s) and are preferentially taken up by cells of the same origin, making them promising candidates for personalized tumor-targeted therapies [5]. Microvesicles (MVs) and exosomes (EXOs) are the two main classes of EVs [6]. Beyond size differences, MVs are formed by outward budding and fission of the plasma membrane, whereas EXOs are released from multivesicular bodies (MVBs) following fusion with the plasma membrane. Upon release, EVs carry a wide variety of biomolecules—including surface proteins, enzymes, microRNAs, proteins, lipids, RNA transcripts, and DNA fragments—originating from the donor cells, which are essential for intercellular communication [7]. These molecular cargos enable targeted interaction with recipient cells. Due to their biological origin, EVs are considered non-toxic, low-immunogenic, and more biocompatible, with the capacity to target specific cell types, making them suitable for cancer drug delivery [8]. EVs are internalized by various mechanisms, including endocytosis, macropinocytosis, phagocytosis, and lipid raft-mediated uptake, with endocytosis being the predominant mechanism in cancer cells [5].

Several studies have explored the use of EVs in cancer treatment, such as loading therapeutic nucleic acids into EVs derived from mesenchymal stem cells (MSCs) [9]. While autologous cancer cell-derived EVs have been shown to enhance cancer cell proliferation [10], their use in targeted cancer therapy remains under investigation. Concerns exist that these EVs may, through intercellular signaling, suppress the immune response and promote a tumor-supportive microenvironment, thus accelerating tumor growth and metastasis [11,12].

Beyond their natural cargo, EVs can be intentionally loaded with specific therapeutic agents such as anticancer drugs [13], short RNAs [14,15], CRISPR/Cas9 [6], and various other therapeutic payloads. Two major strategies are used for EVs loading: cell-based (endogenous) and non-cell-based (exogenous) methods. In cell-based loading, the donor cells are loaded with the therapeutic cargo (e.g., via transfection or incubation), and the EVs naturally carry this material upon release. In non-cell-based loading, EVs are isolated first, and cargo is introduced using methods such as sonication or electroporation [16]. For nucleic acids, electroporation [17] and transfection [14] are commonly employed; for drugs, sonication and incubation are standard techniques [18,19]. Due to the limited understanding of EVs stability under varying environmental conditions [6], cell-based loading is generally considered more effective for ensuring cargo incorporation.

Gold nanoparticles (AuNPs) are representative safe and biocompatible carriers that facilitate cytoplasmic transfection of conjugated nucleic acids [20]. AuNPs can be synthesized via biological or non-biological methods. Biological synthesis (intracellular or extracellular) employs organisms such as bacteria, fungi, and plants [21]. In intracellular synthesis, nanoparticles are reduced inside cells by enzymatic activity, whereas in extracellular synthesis, reduction occurs outside the cells via secreted reducing agents. The extracellular method is considered safer and practical, as it eliminates the need for nanoparticle extraction from cells [22].

The fungus *Fusarium oxysporum* has demonstrated high efficiency in the extracellular production of silver (Ag) and gold (Au) nanoparticles, as shown in our previous work [23,24,25,26,27,28]. Specifically, using the bioproduced AuNPs as NA carriers is attractive due to their unique surface decoration that protects NA cargo and promotes effective transfection. While many characteristics of biologically synthesized AuNPs are known, little is understood about their intracellular fate or whether they can transfer therapeutic NAs via EVs in in vitro models and in vivo conditions.

Regarding non-biologically produced AuNPs, there is a report about tracking EVs loaded with non-biological AuNPs [29]. These AuNPs, depending on their size, can alter the biophysical properties and protein profiles of EVs, and have been shown to suppress cancer cell migration by downregulating cofilin, an actin-binding protein [30]. Moreover, certain non-biological nanoparticles accumulate inside cells without effective exocytosis [31]. For instance, no exocytosis was observed for 20 nm DNA-coated AuNPs after endocytosis into endothelial C166 cells [32].

To reveal the possibility of loading EVs with biological AuNPs carrying NAs and aiming such EVs for parental tumor cells was the main goal of our study. We tested the following: (1) AuNPs loading into EVs and their characterization regarding efficacy and protein content. (2) We verified that the EVs loaded with AuNPs, RNA (antimiR), and transferrin (Tf) are internalized to the parental cell line in vitro. (3) We examined whether the EVs exhibited preferential uptake by the parental cell line compared to another cell line in a co-culture system. (4) We employed a xenograft tumor model to assess the targeting efficacy of autologous EVs containing AuNP-RNA-transferrin conjugates in vivo.

## 2. Materials and Methods

### 2.1. Loading and Characterization of EVs-AuNPs

#### 2.1.1. Biological AuNPs

Gold nanoparticles (AuNPs), produced extracellularly by *F. oxysporum*, were used and methodologies of their preparation and characterization were detailed earlier [33].

In summary, the *F. oxysporum* was cultured in Sabouraud Dextrose Broth (SDB, Sigma-Aldrich, Prague, Czech Republic) at 30 °C for one week. The AuNPs were produced by challenging the cell-free supernatant with HAuCl_4_·3H_2_O (final concentration of 1 mmol; Sigma-Aldrich, Prague, Czech Republic) at 80 °C for 5 min. Washed AuNPs were decorated with Tf and antimiR-135b molecules and characterized with spectrophotometry, Fourier-transform infrared spectroscopy (FTIR), transmission electron microscopy (TEM), energy dispersive X-ray spectroscopy (EDS), zetasizer/DLS, and graphite furnace atomic absorption spectroscopy (GF-AAS).

The AuNPs alone or with cargo exhibited a specific absorption peak at about 530 nm. The average size of the AuNPs alone was 13 ± 1.33 nm and about 50 ± 11.25 nm after conjugation with the Tf and antimiR. The AuNPs as well as conjugates retained their round shape and zeta potential about −36.8 ± 0.45 mV. Concentration of the AuNPs was 11.74 ± 0.02 µg/µL.

On average, the AuNP conjugates contained 0.041 mg/mL of Tf and 0.66 µmol of antimiR-135b per 1 mg of the AuNPs. Successful cargo conjugations were proved with FTIR as well as qPCR (in the case of antimiR-135b) and with a liquid chromatography–mass spectrometry system (LC-MS; in the case of Tf) [33].

#### 2.1.2. Cell Viability Test

The AuNPs were sterilized by tyndallization and examined for toxicity to 4T1 and MDA-MB-231 cells using the MTT assay. 4T1 (ATCC CRL-2539) breast cancer cells were cultured in Roswell Park Memorial Institute Medium 1640 (RPMI-1640, Sigma Aldrich, Prague, Czech Republic), supplemented with 10% fetal bovine serum (FBS, Gibco, Waltham, MA, USA), 4500 mg/L glucose (Sigma Aldrich), 1 mM sodium pyruvate, and 50 mg/mL gentamicin. MDA-MB-231 cells (ATCC HTB-26) were cultured in Dulbecco’s Modified Eagle Medium (DMEM, Sigma Aldrich) with 10% FBS, 4500 mg/L glucose, and 44 µg/mL gentamicin. Cells were seeded in 96-well plates and incubated with serial dilutions of AuNPs (0.5 µg/µL) for 24 h. Subsequently, 3-(4,5-dimethylthiazol-2-yl)-2,5-diphenyltetrazolium bromide (5 mg/mL MTT, EMD Millipore, San Diego, CA, USA) in phosphate-buffered saline (PBS) was added and incubated for 4 h. Color development solution (isopropanol with 0.04 N HCl) was added, and absorbance was measured at 570 nm with a reference at 630 nm using the Infinite 200 PRO UV-visible spectrophotometer (Tecan, Männedorf, Switzerland). Cell viability (%) was calculated as follows:[(A570 − A630) of test cells/(A570 − A630) of control cells] × 100.

#### 2.1.3. EV Isolation

The 4T1 cells were grown in FBS-free conditions using RPMI medium, 10% XerumFree XF212 supplement (Amsbio, Alkmaar, Netherlands), 4500 mg/L glucose, and 1 mM sodium pyruvate. Cells were adapted by gradually replacing the standard medium with the serum-free medium. Once adapted, cells at 80% confluency were treated with a non-toxic dose of AuNPs in the serum-free medium for 5 h. A control group underwent the same procedure without AuNPs. Supernatants were collected and cells were washed with PBS and then incubated overnight with fresh serum-free medium. Final supernatants were harvested, and EVs were isolated using the exoEasy Maxi Kit (Qiagen, Prague, Czech Republic), following the manufacturer’s instructions. EVs were stored at −80 °C in the elution buffer [34].

#### 2.1.4. Internalization and Exocytosis of the AuNPs

The amount of AuNPs internalized during 5 h of incubation was determined in the harvested 4T1 cell supernatant using GF-AAS, following our previous method [26]. For this purpose, 59 µg of AuNPs in 2 mL of the medium was added to the 80% confluent cell plates. Triplicates of samples were incubated for 5 h in cell culture conditions, and supernatants were collected. The surface of the cells was washed with PBS three times to remove surface-bound AuNPs, and this wash was added to the same collection tubes as supernatants. The percentage of uptake was evaluated by calculating the amount of AuNPs that remained outside the cells and the amount of AuNPs in the initial sample.

The cells were then incubated with 2 mL of fresh serum-free medium overnight. The supernatant was harvested for EV isolation (see above), and the cells were washed three times using PBS. The amount of AuNPs was assessed by a GF-AAS. The amount of exocytosed AuNPs was calculated as follows: (AuNPs internalized during 5 h) − (AuNPs remaining inside cells after 24 h).

#### 2.1.5. EV Size, Concentration, and Shape

Zetasizer Ultra (Malvern Panalytical, Malvern, UK) was used to determine the size of the EVs. An ultra-low volume ZEN2112 quartz cuvette containing 40 µL of each sample was used, and the parameters were as follows: water as dispersant, angle of detection-backscatter, temperature 25 °C, and equilibration time 120 s.

The amounts (i.e., concentrations) of the EVs were quantified based on the protein content of the sample using the Bradford assay. Bovine serum albumin (BSA, Sigma Aldrich, Prague, Czech Republic) was used as a reference. A serial dilution of BSA, control EVs, and EV-AuNPs was prepared (range: 12.5 mg/mL to 9.76 µg/mL) and mixed with a Pierce Coomassie Plus (Bradford) Protein Assay Reagent (Thermo Fisher Scientific, Prague, Czech Republic) in a ratio of 1:10. The optical density (OD) of the samples was determined at 595 nm using the Infinite 200 PRO UV-visible spectrophotometry. The standard curve was prepared based on the known amount of BSA, and the protein amount of the sample was determined using the obtained equation.

Shapes of EVs were observed using transmission electron microscopy (TEM). A 5 nm thick carbon layer was deposited on top of 400 mesh copper grids. The grids were then glow discharged and placed on a sheet of parafilm. About 4 µL of the sample was added on top and left to sediment (stick to the carbon layer) for 20 min. Afterwards, the grids were picked up and placed on drops of 0.5% formaldehyde in milliQ water for another 20 min. Then, after 4 quick washes in 4 subsequent drops of milliQ water, the grids were placed on drops of 0.4% uranyl acetate in 2% methylcellulose for 10 min. Afterwards, the grids were picked up, leftover solution was removed with filter paper, and the grids were air dried [35].

#### 2.1.6. Proteome Analysis of EVs

Sodium dodecyl-sulfate polyacrylamide gel electrophoresis (SDS-PAGE) and liquid chromatography with mass spectrometry (LC–MS) analyses were employed to reveal the differences in the protein composition of the EVs before and after loading with AuNPs.

SDS-PAGE was performed according to our previous study [27]. Samples were first denatured in NuPAGE LDS Sample Buffer (Thermo Fisher Scientific, Prague, Czech Republic) at 100 °C for 10 min. Then, they were loaded onto a NuPAGE 10% Bis-Tris Gel (Invitrogen, Thermo Fisher Scientific, Prague, Czech Republic) along with a PageRuler Prestained Protein Ladder (Thermo Fisher Scientific, Prague, Czech Republic). Electrophoresis ran in 1× NuPAGE MOPS SDS running buffer (Thermo Fisher Scientific, Prague, Czech Republic), and then SimplyBlue SafeStain (Thermo Fisher Scientific, Prague, Czech Republic) was used to visualize the protein bands.

LC-MS analysis was also performed according to our previous study [27]. Three replicates of the EVs and EVs-AuNPs were prepared, and after sample pretreatment (alkylation and trypsinization), a liquid chromatography system (Agilent 1200 series, Agilent Technologies, Santa Clara, CA, USA) equipped with a timsToF Pro PASEF mass spectrometer with CaptiveSpray (Bruker Daltonics, Billerica, MA, USA) was used. The proteomics data were obtained using PEAKS Studio 10.0 software (Bioinformatics Solutions, Waterloo, ON, Canada). The results were matched with the UniProt database (all taxa, 11/2021), and the differences between the proteins were determined with Perseus software v2.1.5 and represented as Volcano plots. Because there are some proteins belonging to the capping agents of the AuNPs [27], the protein content of AuNPs as the control was determined in parallel and subtracted from the obtained protein pool of EVs and EVs-AuNPs.

### 2.2. In Vitro Uptake of Loaded EVs by the EV-Parental Cell Line

Before the internalization assay of the EVs-AuNPs, different AuNP conjugates were prepared: AuNPs-antimiR-135b and AuNPs-Tf-antimiR-135b (Tf, Transferrin, human serum, Invitrogen, Thermo Fisher Scientific, Prague, Czech Republic), as previously described [15]. Briefly, antimiR-135b (5′-rUrCrA rCrArU rArGrG rArArU rGrArA rArArG rCrCrA rUrA-3′, 100 µmol) was dissolved at the final concentration of 10 µmol in colloidal AuNP (3.75 ± 0.2 µg/µL)/RNase-free ddH_2_O. The conjugate was incubated in a thermomixer (Eppendorf, Hamburg, Germany) at 1000 rpm at 4 °C overnight and then washed three times with RNase-free ddH_2_O. The dissolved pellet in RNase-free ddH_2_O was used for spectrophotometry and agarose gel electrophoresis studies. A portion was also utilized for the preparation of EVs containing AuNPs-antimiR-135b. To analyze whether the free RNA was washed away, the amounts of RNA in the supernatant from the final step of centrifugation were measured using a Nanodrop spectrophotometer (Thermo Fisher Scientific, Prague, Czech Republic). AuNPs-Tf-antimiR-135b was prepared using the same protocol, but 0.6 µL of diluted Tf in ddH2O (0.5 mg/mL) was added to the mixture, and the conjugate was incubated at 4 °C overnight under shaking conditions. The sample was then washed three times using RNase-free ddH_2_O, and the dissolved pellet was used for spectrophotometry and LC-MS analysis [33]. The same protocol for evaluating free RNA was followed, as described above.

For cell internalization assay, the EVs, AuNPs-antimiR-135b, and AuNPs-Tf-antimiR-135b were incubated with 4T1 cells (at their non-toxic doses according to MTT assay result), and then EVs were extracted. In parallel, 10 plates of 4T1 cell line were cultured, and the test and control samples were added to each plate: (1) the conditioned 4T1 cell supernatant after addition of AuNPs-antimiR-135b and before EV extraction, (2) the conditioned cell supernatant after addition of AuNPs-Tf-antimiR-135b and before EV extraction, (3) the conditioned cell supernatant that was used for control (unloaded) EV extraction, (4) the control cells without any treatment, (5) the elution buffer that is used for the final step of the EV extraction, (6) EVs containing AuNPs-Tf-antimiR-135b, (7) EVs containing AuNPs-antimiR-135b, (8) extracted EVs from the control cells, (9) non-toxic dose of AuNPs-Tf-antimiR-135b, and (10) non-toxic dose of AuNPs-antimiR-135b. After overnight incubation, the cells were washed three times with PBS, and micro-RNA was extracted using the High Pure miRNA isolation Kit (Roche, Prague, Czech Republic). The amounts of the extracted RNA were determined using a spectrophotometer, and cDNA was prepared using the High-Capacity cDNA Reverse Transcription Kit (Thermo Fisher Scientific, Prague, Czech Republic). The used primers were miR-135b (assay MI0000810) and miR-16 as an internal control (assay MI0000070) (Thermo Fisher Scientific, Prague, Czech Republic). TaqMan Universal PCR Master Mix (No AmpErase) and primers (miR-135b assay MI0000810 and miR-16 assay MI0000070) were used to perform the real-time polymerase chain reaction (qPCR) in an iQ5 Real Time PCR Detection System (BioRad, Prague, Czech Republic), and the iQ5 Optical System Software 2.1 (BioRad, Prague, Czech Republic) was used to interpret the resulting data [20]. The level of miR-135b was normalized to internal control miR-16 and the change in miR-135b level was estimated based on the standard 2^−ddCt^ method [36]. The cutoff for samples with remarkably decreased miR-135b was fold of change ≤0.5. Statistical significance was revealed based on data from three experiments, presented as averages and standard deviations. Values of *p* ≤ 0.01 (**) were considered statistically significant.

### 2.3. EVs Tropism Assessment In Vitro

To check the tropism of the EVs containing the AuNP conjugate, the cellular co-culture system of two cell types without cell-to-cell contact with the transwell inserts and a permeable membrane (0.4 µm pore size, life science, Prague, Czech Republic) was used. Four inserts were cultured by 4T1 cells in the working medium consisting of RPMI 1640. Four wells in a 12 well plate were cultured by the MDA-MB-231 cell line in the working medium of DMEM. After 24 h incubation, when both cell lines reached 80% confluency, media from the inserts and wells were removed, and the insert compartments were placed in wells and filled with 2 and 4 mL of each corresponding medium, respectively. One well was considered as a control and remained without addition of EVs or conjugate, and the other wells had the AuNPs-Tf-antimiR-135b conjugate, EVs containing AuNPs-Tf-antimiR-135b conjugate, and empty EVs. In the case of EVs, 50 and 100 µL of the EVs were applied onto the inserts and wells, respectively. In the case of conjugate, 2.5 and 5 µL of the AuNPs-Tf-antimiR-135b conjugate were added to the insert and well, respectively. The incubation of the cells continued for 24 h and the next day the cells from inserts and wells were washed three times with PBS, harvested, and used for microRNA extraction and qPCR with the same method described above.

### 2.4. Application of Loaded EVs In Vivo

#### 2.4.1. Mice Model and Obtaining Tissue Samples

The animal experiments were approved by The Ethical Committee for Animal Experimentation of the CAS under ID 90-2024-P. Twenty female BALB/c mice aged 8–12 weeks were injected subcutaneously into the fourth right fat pad of the mammary gland with 10^6^ freshly prepared 4T1 cells in PBS (in a volume of 0.1 mL). When the tumor diameter reached 5 mm in size after 10 days, the first dose of the samples was administered. The animals received one dose of the samples. There were four groups, each containing 5 mice and selected randomly: PBS (negative control), empty EVs (negative control), EVs containing AuNPs-Tf-antimiR (test group), and AuNPs-Tf-antimiR (positive control).

The samples were administered through tail vein injection and the dosage was determined through the MTT assay (the dose used was one dose lower than the IC_50_). The volume of the control and the test samples was consistent for all animals (20 µL). Five days after the administration of the treatments, the animals were sacrificed by cervical dislocation under anesthesia (ketamine/xylazine). Tumors were extracted and their volumes were determined according to the formula V = $\frac{L\;\times \;S^{2}}{2}$ (where V is the tumor volume, *L* is the longest tumor axis (mm), and *S* is the shortest tumor axis (mm)). ANOVA/post hoc Tukey test for group comparison (online tool available at https://astatsa.com/ accessed on 31 July 2025) was used and *p*-value < 0.05 was considered as statistically significant.

Tumor tissues then were sectioned into four parts; one part was cut into small cubes (1 mm^3^) and placed in a cold and fresh fixator and transferred immediately for TEM analysis. The other sectioned tissue was placed in the freshly prepared 4% formaldehyde and stored at 4 °C for microscopic analysis after hematoxylin and eosin staining (H & E staining). The third part was homogenized using PBS and a mechanical technique and the cells were fixed in 70% ethanol at −20 °C for flow cytometry analysis. The fourth part was homogenized using a 20% binding buffer of the microRNA extraction kit and RNA was extracted according to the protocol as described later in this manuscript.

#### 2.4.2. TEM and EDS Analyses

The fresh fixator used in this study consisted of 3% formaldehyde and 1% glutaraldehyde in 0.1 M sodium cacodylate buffer (SB, pH 7.4). The samples were preserved at 4 °C overnight and washed in the SB buffer and fixed in 1% OsO_4_ in SB. After washing with SB and ddH_2_O, dehydration using serial concentrations of acetone in ddH_2_O, Epon Araldite resin embedding was performed. The prepared sections were analyzed under the Jeol JEM-F200 TEM (JEOL Ltd., Tokyo, Japan; operated at 200 kV) that was equipped with the TVIPS XF 416 CMOS camera (TVIPS GmbH, Gilching, Germany). The sections were analyzed by a JED 2300 X-ray spectrometer (JEOL, Ltd.) to obtain EDS data and analyzed by Jeol AnalysisStation for JEM-F200.

#### 2.4.3. H & E Staining

Paraffin embedded blocks of the tumor tissues were prepared and sectioned into 5 μm thickness using Leica RM2255 Fully Automated Rotary Microtome (Leica Biosystems, Deer Park, IL, USA). The slides were stained by the H & E method and parameters such as expansion of neoplastic areas, expansion of necrotic and apoptotic areas, increased hyperemia, and increased inflammatory cells in different groups were checked and compared together using a light microscope.

#### 2.4.4. Flow Cytometry

The ethanol-fixed cells were washed twice with PBS at 500× *g* for 8 min. The pellet was resuspended in 0.1% Triton X-100 and incubated at 4 °C for 3 min. The samples were then centrifuged at 500× *g* for 8 min, resuspended in RNase (100 units/mL), and incubated at room temperature for 10 min. Following centrifugation, the pellet was homogenized in PI (50 µg/mL) and incubated at 4 °C for 1 h in the dark. The samples were analyzed using a NovoCyte Quanteon flow cytometer (Agilent Technologies, Inc., Santa Clara, CA, USA) and the Cell Cycle Analysis Tool. Fluorophore settings for PI detection were set automatically and were based on calibration with QC beads before each experiment. The results were compared together and statistically analyzed using the post hoc Tukey HSD test calculator with Scheffe, Bonferroni, and Holm multiple comparison (available at https://astatsa.com/ accessed on 31 July 2025), and *p*-value < 0.05 was considered as statistically significant.

#### 2.4.5. qPCR Analysis

RNA was extracted from homogenized samples by the same methods described in Section 2.2. of this manuscript. The qPCR analysis and normalization were also performed as described above using the same primers for the miR-135b and the miR 16 detection; the change in miR-135b level was estimated based on the standard 2^−ddCt^ method and remarkable decrease in miR-135b level was set as fold of change ≤0.5.

## 3. Results and Discussion

### 3.1. Loading and Characterization of EVs-AuNPs

Initially, parental 4T1 cells were cultured in XerumFree XF212 medium without FBS. However, we experienced changes in cell morphology (apoptotic cells) and the low growth rate that was also observed by other researchers [16,21]. To address this, a sequential adaptation (Weaning method) of the cells was performed by gradually increasing the concentration of the XerumFree XF212 medium supplement. Additionally, the cells were cultured in antibiotic-free conditions to avoid the impact of antibiotics on cell growth [37]. The parental (4T1) as well as non-parental (MDA-MB-231) cells used later in the study were tested for potential toxicity of the AuNPs. EVs isolated from parental 4T1 cells incubated with the AuNPs were characterized for AuNPs presence, AuNPs load efficacy, or changes in protein composition of the loaded vs. empty EVs.

#### 3.1.1. Cell Viability Test

The cytotoxicity of the AuNPs against 4T1 cells was assessed prior to EV extraction to determine the optimal sub-IC_50_ dose. The assessment of the AuNPs-mediated cytotoxicity in MDA-MB-231 cells had two aims: first, to assess and compare the cytotoxicity of AuNPs in the parental and non-parental cell lines, and second, to differentiate potential cytotoxicity of the AuNPs vs. non-parental EVs while using EVs loaded with the AuNPs, as described later in this article.

Figure 1 shows the cell viability percentage of the cells at different concentrations of AuNPs after 24 h of incubation. Based on the MTT assay results, the IC_50_ concentration of AuNPs for 4T1 cells was 0.25 µg/µL. To avoid compromising cell viability due to high AuNPs concentrations, we treated the 4T1 cell culture with 0.03 µg/µL of AuNPs to obtain EVs-AuNPs. Testing the AuNPs-mediated toxicity in the two different cell lines showed no (in lower concentrations) or a very small difference (higher concentrations) between the two cell lines.

#### 3.1.2. Internalization and Exocytosis of the AuNPs

The 4T1 cells were sequentially adapted to grow well in the XerumFree XF212 medium without FBS and antibiotics. To determine the amounts of AuNPs that were internalized by cells and then exocytosed inside the EVs, three sample sets were incubated with diluted initial amounts of AuNPs (58.7 µg/2 mL). After 5 h of incubation, the remaining Au amounts outside the cells were determined using GF-AAS. The amounts of AuNPs in the initial sample, as well as the AuNPs that remained inside the cells after overnight incubation, were also analyzed using GF-AAS. The amounts of AuNPs that were internalized and then exocytosed inside the EVs were determined and listed in Table 1.

Based on the information from Table 1, the calculated total amount of AuNPs inside the EVs after incubation with AuNPs was 38.64 ± 0.29 µg. The efficacy of the loading was then about 66%.

#### 3.1.3. EV Size, Concentration, and Shape

Two types of EV populations were observed, one around 60 nm (with intensity less than 10%) and the other around 250 nm (with the intensity around 90%). Further assessment with TEM was necessary to confirm that both populations were EVs. Table 2 and Figure 2 present the results of the size measurements.

The size of isolated empty (control) EVs as well as EVs loaded with the AuNPs corresponds to the microvesicle subpopulation of EVs derived from the 4T1 cells [38]. According to Kang et al. [39], EVs can be broadly categorized based on their size: small EVs are less than 100 nm, while large EVs are typically greater than 200 nm. Our findings indicated that the majority of the EVs were classified as large EVs. This raises a question about whether different sizes of EVs could impact their biodistribution and the tumor-targeting ability of the nanoparticles. The review by Kang et al. reports that both small and large EVs are found at tumor sites. Specifically, small EVs were detectable in tumors between 2 and 12 h, as well as at 24 h post-administration. In contrast, large EVs were only detectable within the 2 to 12 h window after injection. Tumors often have a leakier vasculature compared to healthy blood vessels and impaired lymphatic drainage, which facilitates easier passage of nanoparticles into the tumor interstitium and results in higher retention within tumor tissue. Since EVs are comparable in size to liposomes and other nanoparticles, it is plausible that their accumulation in tumors occurs through similar mechanisms. Therefore, the EV size does not appear to significantly influence their biodistribution. Also, the size of EVs probably will not affect the tumor-targeting capability of the AuNPs. This statement is supported by our findings that the recognition of the target cells is determined mainly by the EVs tropism and the targeting molecules on the AuNP surface may play a role in higher EV uptake into parental cells only (see data below).

#### 3.1.4. Encapsulation of the AuNPs into EVs

TEM images obtained from both control EVs and EV-AuNPs proved encapsulation of AuNPs into EVs, and representative images are shown in Figure 3.

As represented in Figure 3, some EVs were observed in a cup shape due to collapse of their membrane during the TEM procedure [40]. Some EVs exhibited a dark color in TEM images, which is in agreement with observations of other groups reported earlier [41]. Since the presence of AuNPs cannot be distinguished in the dark EVs, data for this type of EV-AuNPs are not included. To confirm the presence of AuNPs inside the dark EVs, EDS analysis was performed (Figure 4).

Our findings confirmed the presence of gold nanoparticles (AuNPs) within extracellular vesicles (EVs). Previous research has shown that both healthy and diseased cells release EVs composed of a lipid bilayer that contain various biomolecules, including cell surface antigens, intracellular proteins, proteases, microRNAs, and messenger RNAs derived from the parental cells. The composition of these EVs varies depending on their parental and target cells, enabling communication between cells through the transfer of signals [40].

Furthermore, multiple research groups have established techniques to generate EVs loaded with nanoparticles. The most widely used method involves incubating cells directly with media containing synthetic nanoparticles, which are then internalized and later secreted within exosomes via the exosomal biogenesis pathway. We expect that, in our experiment, following incubation with AuNPs, the nanoparticles were incorporated into EVs through a similar internalization process [41].

Unpublished preliminary data from our laboratory suggest that caveolae-mediated endocytosis and macropinocytosis are the primary mechanisms for AuNP uptake. Our results indicate that when clathrin-dependent endocytosis is unblocked and other pathways are inhibited, AuNPs mainly enter cells via clathrin-mediated endocytosis, eventually being degraded in lysosomes. Transmission electron microscopy (TEM) images demonstrated that AuNPs were localized in early and late endosomes and later secreted within EVs, supporting their intracellular trafficking and export (unpublished data). Additionally, electron microscopy studies showed that early endosomes generally contain few or no intraluminal vesicles (ILVs), whereas late endosomes often contain between three and twenty ILVs. It remains unclear whether the formation pathway of ILVs influences whether vesicles are directed towards lysosomal degradation or towards the plasma membrane to be released as EVs [42].

In studies examining the uptake and drug delivery capabilities of EVs, most employ fluorescent labeling techniques such as lipid dyes or fluorescently tagged proteins. These methods have demonstrated that EVs can effectively transfer proteins and mRNA between donor and recipient tumor cells both in laboratory and experimental animals. Notably, EVs facilitate communication within a tumor and between distant tumors, contributing to tumor migration and metastasis. It has been observed that, following systemic diffusion, both small and large EVs deliver their cargo through endocytosis [43], a process relevant to our research.

#### 3.1.5. Proteome Analysis of EVs

The Bradford assay was used to roughly detect any protein in EVs and EV-AuNPs. The amounts of proteins were determined using an equation based on the standard curve of known amounts of BSA shown in Appendix A. The protein content in the control EVs (empty EVs) was 25.4 µg/mL, in contrast to the protein load in the EV-AuNPs, which was 45.5 µg/mL. The higher protein content in EV-AuNPs may be due to the presence of AuNP capping agents. This hypothesis was further confirmed by SDS-PAGE and LC-MS.

Both EVs and EV-AuNPs were analyzed by SDS-PAGE to evaluate basic differences in protein composition present in the two groups. As presented in Figure 5, there are some common protein bands in both groups, including two around 115 kD, one with a size of 50 kD, and one with a size of 15 kD. The concentration of the 50 kD protein in EV-AuNPs appeared to be higher than in control EVs. To reveal the protein composition, we used LC-MS analysis.

The LC-MS analysis of control EVs and EVs loaded with AuNPs (EV-AuNPs) revealed common proteins between both samples, as well as some proteins that were unique to each sample. AuNPs were used as a control in this step because they carry capping proteins on their surface, which interact with functional groups and different pHs. Our previous study revealed some of these proteins [38]. After subtracting the proteins detected in the control (AuNPs only) sample, LC-MS results identified 222 proteins, among which 12 proteins were significantly changed between tested groups (i.e., EV-AuNPs vs. EVs). The composition of certain proteins in the EV-AuNPs group differed from the control EV group, indicating that the addition of AuNPs to the 4T1 cell culture altered the protein profile of the EVs. Figure 6 displays the Volcano plot and the specific proteins in each group, while Table 3 lists the proteins mentioned in Figure 6.

Individual roles of the identified proteins were retrieved from the UniProt database (www.uniprot.org). Based on these data, we conclude that most of the proteins are involved in extracellular matrix–cell linking or cell cytoskeleton maintenance (i.e., alpha-actinin-4, laminin subunit beta-1, collagen alpha-2(IV) chain, bone morphogenetic protein 1, carboxypeptidase E, and stress-induced-phosphoprotein 1). Their upregulation probably reflects changes in adhesion and cell–matrix binding after AuNPs uptake and processing. Ceruloplasmin manages Fe ion transport across membranes and its upregulation likely reacts to the transferrin presence on the AuNP surface. Involvement of such proteins in response to uptake of AuNP-loaded EVs was expected. The small nuclear ribonucleoprotein-associated protein B and N and the GTP-binding nuclear protein Ran are involved in RNA processing and transport and their possible response to AuNP (or the carried antimir) needs to be further elucidated.

The two important findings, i.e., (1) EVs loaded with AuNPs exhibited higher protein content, likely originating in the AuNPs’ capping agent, and (2) the proteome differs between empty EVs and EVs loaded with the AuNPs, agree with published observations of other groups [30,43] and support the hypothesis that biologically produced as well as non-biologically produced AuNPs alter the proteome of interacting cells.

### 3.2. In Vitro Uptake of Loaded EVs by the EV-Parental Cell Line

#### 3.2.1. Identification of AuNP-Tf Within EVs

The AuNPs were decorated with Tf of human origin in order to facilitate cancer cell targeting. The cross reactivity with the mouse Tf receptor is sufficient for conjugate internalization by the mouse tumor model too. Using a Tf of different origin than the recipient cancer cells supports further identification of the AuNP conjugates within the tumor-derived EVs. Presence of human Tf was detected by LC-MS in the EVs loaded with AuNPs-Tf-antimiR 135b (Table 4).

#### 3.2.2. Reduction of miR-135b Revealed by qPCR

AntimiR-135b represents effective ways to reduce the miR-135b level in tumor cells. Its effector action proves that the AuNP carriers entered the target cell cytoplasm and released the cargo in order to act. We analyzed the inhibitory effect of EV-AuNPs-antimiR-135b and EV-AuNPs-Tf-antimiR 135b on the mir-135b target after incubation of the 4T1 cell line with differently loaded EVs (Figure 7).

The 4T1 cells were incubated with non-toxic doses of AuNPs-antimiR 135b with or without Tf. EVs were then extracted from the untreated control cells (“empty” EVs), EVs containing AuNPs-antimiR 135b, and EVs containing AuNPs-Tf-antimiR-135b. The non-treated cells served as negative controls for evaluation of cells treated directly with functionalized AuNPs. The 4T1 cells treated with empty EVs served as negative controls for evaluation of cells treated with AuNPs carriers encapsulated into EVs. Furthermore, we also tested the effect of conditioned acellular media collected from cell culture after EV production and before their extraction. These data are shown in Appendix A.

As expected, the cells incubated with AuNPs-antimiR 135b with or without Tf exhibited significantly reduced levels of miR-135b (*p*-values < 0.01 in both samples). The level of miR-135b in cells stimulated with empty EVs was a little bit higher to cells that remained untreated (non-significant change, *p*-value = 0.9). Since miR-135b can be naturally transported by EVs among the producing cancer cells, the increase in miR-135b level likely reflects the miR-135b produced by stimulated cells together with miR-135b obtained by the cells within EVs. After incubation of 4T1 cells with EVs-AuNPs-antimiR 135b with or without Tf, we observed a decrease in miR-135b in both samples (*p*-values < 0.01 in both samples, *p* = 0.001 and *p* = 0.006, respectively). However, the miR-135b level in cells incubated with EVs-AuNPs-antimiR-135b without Tf remained above the 0.5 cutoff. This observation suggests that Tf decoration of AuNP carriers played an important role when the conjugate was transported among cancer cells via EVs.

In our previous studies, we employed Tf to facilitate uptake of the AuNPs into target tumor cells due to high expression of the Tf receptor (TfR) on most cancer cells [44]. As presented in Figure 7, in 4T1 cells, specifically, the Tf did not probably further promote the uptake of the AuNPs carrying antimiR and the efficacy of miR-135b silencing was comparable in complexes with or without the Tf. On the other hand, once the AuNPs-antimiR were loaded into the EVs, the presence of Tf on the AuNPs surface increased the efficacy of the miR-135b silencing by 38%. These data agree with reports regarding Tf-facilitated nanoparticle internalization and endocytosis [36]. Tf binds to iron and attaches to the TfR, leading to clathrin-mediated endocytosis and transporting iron into the cells [44]. The hypothesis that Tf actively helps with AuNP endocytosis into the cells was also supported with output of the LC-MS analysis, confirming the presence of Tf in EVs loaded with AuNPs-Tf-antimiR 135b.

The level of miR-135b silencing in vitro is more prominent in the cells treated with AuNPs-antimiR alone than after their loading into vesicles. This finding was not expected; we suggest there were several points that could affect the output, and we will focus on them in our following studies. First, the AuNPs-antimiR alone are smaller in size than the EVs-AuNPs-antimiR and even though there is no Tf on their surface, there is still a capping agent originating from biological production of the AuNPs. The capping agent can promote uptake of the carriers with their cargo [20,33]. Within EVs, the capping agent is masked by the vesicle membrane and cannot be involved in interactions with the target cells. Second, the amount of EVs-transported antimiR was lower due to additional steps such as loading EVs with AuNPs-antimiR. Also, the antimiR could be released from the AuNPs within the EVs, for example, in response to a change in pH, and degraded or lost during the transport to cytoplasm of recipient cells. This finding opened many questions to be addressed in the next project.

### 3.3. EVs Tropism Assessment In Vitro

In order to assess the tropism of EVs extracted from the 4T1 cells, we employed a co-culture of two cell lines 4T1 and MDA-MB-231. Based on results demonstrated in Figure 7, we loaded the 4T1-derived EVs only with the AuNPs-Tf-antimiR. The effect of the AuNPs-Tf-antimiR on the target miR-135b in both cell lines in co-culture was checked using qPCR (Figure 8).

As shown in Figure 8, the two cell lines possessed different base levels of the target miR-135b as the level of miR-135b in cells stimulated with empty EVs is higher in the MDA-MB-231 cells than in 4T1 cells, but the AuNPs-Tf-antimiR-135b acted effectively in both the cell lines. After internalization of the AuNPs-Tf-antimiR-135b into the EVs derived from 4T1 cells, the efficacy significantly shifted in favor of the 4T1 cells. While the statistical analysis revealed there was a significant difference in the level of miR-135b in the co-culture of 4T1 and MDA-MB-231 cells after incubation with EVs-AuNPs-Tf-antimiR-135b that were derived from 4T1 cells (*p* value < 0.01), using conventional 2^−ddCt^ algorithm and a cutoff = 0.5, we proved a remarkable decrease in the target miR-135b only in parental 4T1 cells stimulated with EVs-AuNPs-Tf-antimiR-135b, which is in agreement with a previously published report [29].

### 3.4. Application of Loaded EVs In Vivo

Since we confirmed that the 4T1 EVs internalize specifically to their parental cell line in vitro, we wondered if they will home into 4T1-derived tumors in vivo. Our goal was to see if the EVs will enter the tumor cells, transfer the AuNPs-antimiR, and eventually trigger any response such as decreased level of miR-135b or change in cell viability or apoptosis. There were four different groups, including PBS (negative control), empty EVs (negative control), EVs containing AuNPs-Tf-antimiR (test group), and AuNPs-Tf-antimiR (positive control), that we analyzed for tumor growth as well as AuNPs presence, morphology changes of the target tumor tissue, and miR-135b knock-down.

#### 3.4.1. Tumor Growth Assessment

The diameters of the tumors in each group were determined, and the results are shown in Figure 9.

Analysis of tumor sizes among different groups, using one-way ANOVA with post hoc Tukey HSD and Scheffé tests, showed that there were no significant differences between the tested groups (*p*-value > 0.05).

Considering the fact that our effector molecule was the antimiR-135b, not a cytostatic drug, we did not expect remarkable changes in tumor growth. Moreover, the administration of a single dose of antimiR-135b was performed, and the animals were sacrificed after five days. It appears that this procedure may not result in a significant change in tumor size. Autologous cancer cell-derived EVs containing cytostatic drugs like methotrexate or cisplatin were studied earlier and showed increased drug uptake by the specific tumor cells and selective EV accumulation in the tumor site and decreased tumor volume in vitro and in vivo, respectively [45,46].

#### 3.4.2. TEM and EDS Analyses

The presence of the AuNPs within tumor tissues derived from experimental animals was analyzed using TEM and EDS. Representative TEM images of the tested groups are presented in Figure 10, followed by the EDS data analysis (Figure 11) to confirm the presence of Au in the studied sections of the two groups stimulated with the AuNPs.

TEM and EDS analyses of the excised tumor tissues confirmed the presence of AuNPs within the tumor after tail vein administration of AuNPs-Tf-antimiR alone and EVs containing AuNPs-Tf-antimiR. In representative Figure 10 and Figure 11, the AuNPs-Tf-antimiR alone can be seen in the cell cytoplasm and the AuNPs-Tf-antimiR delivered by EVs can be seen in the late endosome. Thus, both complexes—with and without EVs—were successfully delivered by the peripheral blood system into the tumor site. The efficacy of each delivery was further examined by H & E staining and level of microRNA-135b knock-down.

#### 3.4.3. H & E Staining

Tumor samples from four groups underwent analysis by light microscope. The results are presented in Figure 12 and are as follows:

**PBS-treated group:** In the pathological sections of the PBS-treated group, the neoplastic and tumoral areas of the breast are extensive, with neoplastic and mitotic cells widely scattered throughout the tissue. Various stages of mitosis are visible in most of these cells. The characteristics of the tumor cells include undifferentiated morphology with severe polymorphism, irregular and hyperchromatic nuclei, and basophilic cytoplasm. Necrotic areas are generally small, with few necrotic and apoptotic cells, and inflammatory cells are not visible. Hyperemia is developing in the tissue mentioned above (Figure 12(A1–A3)).

**Group treated with AuNPs-Tf-antimiR-135b:** In the pathological sections of the samples from the AuNPs-treated group in the breast, the distribution of the neoplastic area is limited, with necrotic areas and necrotic and apoptotic cells spreading sporadically. Additionally, the number of pyknotic nuclei is increasing. Neoplastic cells exhibiting polymorphism and an undifferentiated state, as well as mitotic cells, are observed in small numbers and only in certain areas of the tumor. Inflammatory cells are present in moderate numbers in some areas of the tumor, and a small amount of blood accumulation is also noted in the tumor regions (Figure 12(B1–B3)).

**Empty EVs-treated Group:** In the pathological sections of empty EVs-treated samples, necrotic areas in the breast tumor region were centrally expanded, while neoplastic areas were limited. Neoplastic cells exhibited polymorphism and had dark, compact nuclei, and the number of cells undergoing mitosis was reduced. Most of the cells in the necrotic area displayed pyknotic nuclei with karyorrhexis and little cytoplasm, indicating an increase in apoptosis. Vascular dilation and blood accumulation were widespread in the tumor areas. Inflammatory cells were observed in moderate and scattered numbers in certain tumor regions (Figure 12(C1–C3)).

**EVs-AuNPs-Tf-antimiR-135b-Treated Group:** In the pathological sections of samples from the EVs-AuNPs-Tf-antimiR-135b-treated group, both neoplastic and necrotic areas in the breast tumor region are equally expanded, with necrotic areas concentrating in the aforementioned tissue. Neoplastic cells are observed in small numbers in most non-tumor areas and exhibit reduced polymorphism. Additionally, a decrease in the process of mitosis is noted in the tumor regions. Cells in the necrotic area often display pyknotic nuclei, karyorrhexis, and scant, unclear cytoplasm. The amount of blood accumulation in the tumor areas is diminished, and inflammatory cells are present in moderate numbers and are scattered throughout the tissue (Figure 12(D1–D3)).

Finally, in order to compare and classify the results, the grading of changes in four main criteria, including expansion of neoplastic areas, expansion of necrotic and apoptotic areas, increased hyperemia, and increased inflammatory cells, is shown in Table 5.

Importantly, even though the tumor volume remained unchanged as discussed above, the histological analysis revealed some changes in the tumor tissue that could be attributed to the effect of antimiR-135b, such as expansion of necrotic and apoptotic areas or increased inflammation. The increased rate of apoptotic cells was further supported by the increased percentage of cells with a sub-G1 peak, as revealed by flow cytometry in Section 3.4.5. The induction of apoptosis by antimiR-135b agrees with reports from our group as well as other groups [47].

#### 3.4.4. qPCR Analysis

Tumor samples, after RNA extraction and cDNA synthesis, were analyzed for differences in their miR-135b levels using qPCR. The internal control was miR-16, and the difference between the levels of miR-135b normalized to miR-16 is shown in Figure 13.

Group comparison ANOVA with the post hoc Tukey HSD test showed no significant differences between any of the test and control groups (*p* > 0.01). On the other hand, the levels of miR-135b in samples that received AuNPs-Tf-antimiR or EVs-AuNPs-Tf-antimiR were lower than the cutoff of 0.5, pointing out a remarkable knock-down of the target miR-135b. The efficacy of miR-135b knock-down is slightly higher in tissues targeted with EVs-AuNPs-Tf-antimiR, although there was a higher standard error originating from the small experimental group.

#### 3.4.5. Flow Cytometry

In the current study, cells that were extracted from the tumors were analyzed for their cell cycle. Figure 14 represents the results from the analysis of the cell cycle between four different groups.

The post hoc Tukey HSD test with Scheffe, Bonferroni, and Holm multiple comparison showed that there was a significant difference in the sub-G1 phase between the PBS and AuNPs-Tf-antimiR-135b groups (Tukey HSD, *p*-value = 0.019 and Scheffe, *p*-value = 0.026), and there was no significant difference between the tested groups in G1, S, and G2 phases (*p*-value > 0.05). The significant difference in the sub-G1 phase between the PBS and AuNPs-Tf-antimiR-135b groups (*p*-value = 0.033) and a significant difference between the PBS and EVs-AuNPs-Tf-antimiR-135b groups (*p*-value = 0.045) were confirmed by Bonferroni and Holm comparison.

Flow cytometry analysis of the cell cycle also revealed a potential decrease in cells in G1 and G2 phases but the data are not homogenous and standard deviations are too high to make a statement. However, there are some reports suggesting that EVs themselves can increase cell proliferation (that would correspond to decrease in G1/G2 phases). For example, EVs derived from human bone marrow mesenchymal stem cells (MSCs) [48] or EVs derived from fibroblast-like mesenchymal cells [49] could increase the viability of cancer cells and tumor growth. Native EVs increased the prostate cancer cell line viability but after loading with a drug, they increased the paclitaxel cytotoxicity and apoptotic effects in LNCaP and PC-3 cells [13]. Similarly, another study reported that while EV-containing AuNPs conjugated with doxorubicin derived from human non-small-cell lung cancer cell line showed selective cytotoxicity to autologous cancer cells in contrast to the fibroblast cells, the EVs without the conjugate increased the autologous cell line cell viability [50]. Among other things, the dual action-boosting cancer cell proliferation of EVs while empty and the boosting anticancer effect of EVs while loaded with cytostatics limit practical use of the EVs. Even though they are prospective tools and possess many advantages, more studies on their safety and efficacy are needed.

Our data analysis showed that, although the application of EVs-AuNPs-Tf-antimiR in vivo did not show a significant difference in tumor sizes among the groups, the EVs containing AuNPs-Tf-antimiR, after tail vein administration, located their target (parental cell line) and exhibited their effects at the molecular level. This was the first study on the application of biologically produced AuNPs in tumor targeting using EVs. Importantly, the EVs successfully delivered AuNPs-Tf-antimiR-135b into parental tumor cells in vivo and the amount of antimiR delivered was high enough to knock-down the level of target miR-135b. This finding supports previously published research regarding the use of EVs containing AuNPs as effective vectors for metastatic lung tumors in vivo [29].

## 4. Conclusions

In summary, we successfully loaded EVs with AuNPs carrying antimiR-135b and we found that the Tf molecule on the AuNPs surface significantly increases the efficacy of EVs-AuNPs-antimiR with respect to silencing the target miR135b. We proved silencing of the oncogenic miR135b in vitro as well as in vivo and we witnessed changes in cell prosperity that could be triggered by the antimiR-135b.

Our findings support the hypothesis that cancer cell-derived autologous EVs can be effectively used as personalized drug delivery vesicles in original cancer treatment. The key advantage of using extracellular vesicles (EVs) as drug delivery systems is their ability to mitigate the toxic responses often associated with foreign substances in the body. Because EVs originate biologically, they tend to elicit minimal immune reactions. Additionally, EVs are considered safe since they do not replicate and are not mutagenic, thereby reducing worries about adverse effects or tumor development. These safety features have been supported by the low toxicity levels reported in in vivo studies involving EV-based therapies [40]. On the other hand, considering the fact that EVs are essentially parts of cancer cells, bearing tumor markers as well as tumor specific molecules, extensive studies are still needed to ensure safety of reintroduction of such cancer-derived EVs back to the treated individuals.

## Figures and Tables

**Figure 1 pharmaceutics-17-01015-f001:**
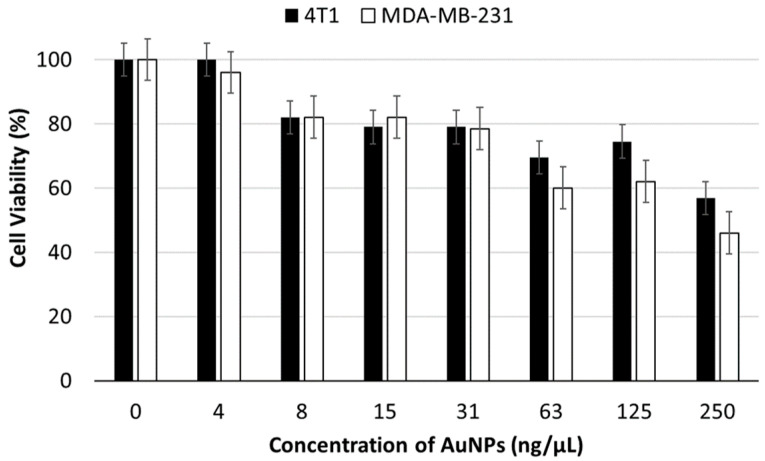
Viability of 4T1 and MDA-MB-231 cancer cells incubated with a serial dilution of AuNPs for 24 h.

**Figure 2 pharmaceutics-17-01015-f002:**
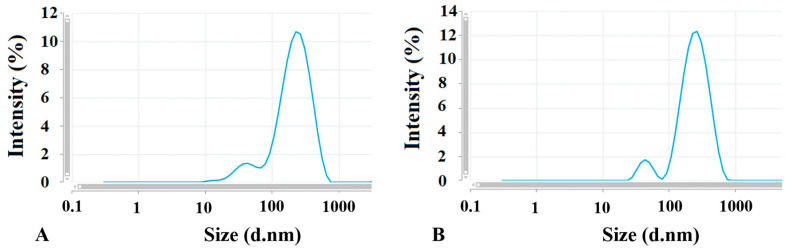
Two peaks suggesting two EV populations defined with different sizes measured by Zetasizer. (**A**) control EVs and (**B**) EV-AuNPs.

**Figure 3 pharmaceutics-17-01015-f003:**
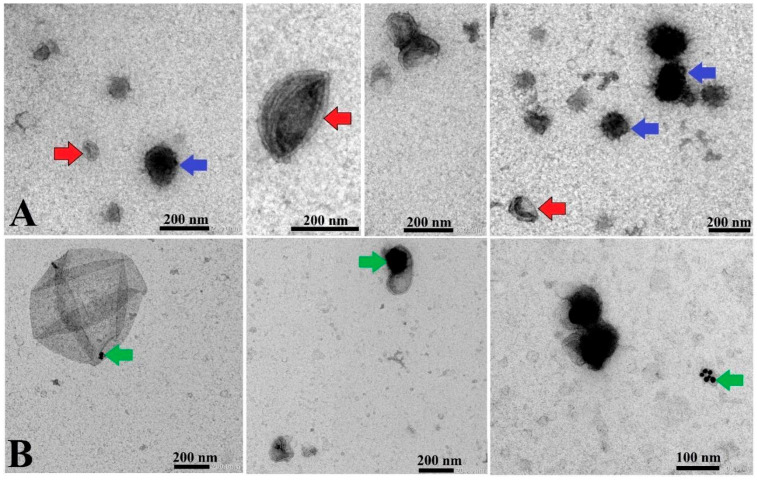
TEM analysis of EVs and EV-AuNPs. (**A**) Control EVs with round shape and double membranes. Red arrows indicate cup-shaped EVs, and blue arrows indicate EVs with darker color. (**B**) EV-AuNPs group, with small AuNPs visible in some EVs (green arrows). Scale bars = 200 nm and 100 nm.

**Figure 4 pharmaceutics-17-01015-f004:**
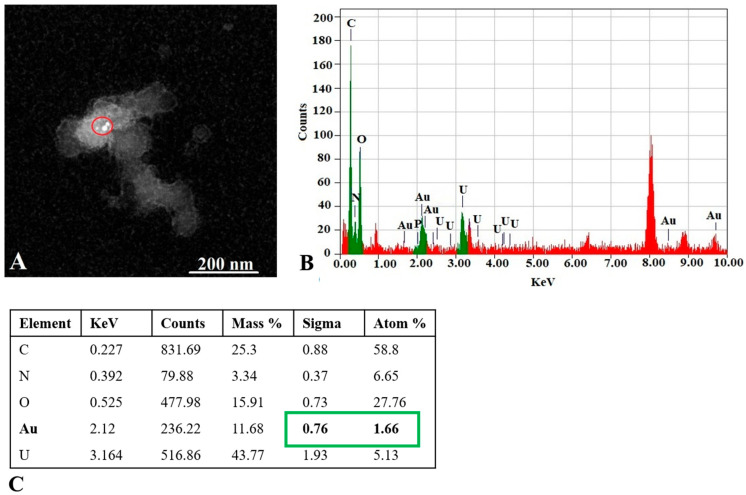
EDS analysis of the EV-AuNPs sample. (**A**) STEM image with selected region of interest (ROI, red circle) of the EV-AuNPs sample. (**B**) EDS graph and elemental analysis of ROI. Au (green) was detected in the ROI. Background is in red. (**C**) Numeric values of peaks detected in the ROI. Au content is marked in green. Atom % = percentage of analyzed element atoms out of all detected atoms in selected region, Sigma = standard deviation (SD) of atom %.

**Figure 5 pharmaceutics-17-01015-f005:**
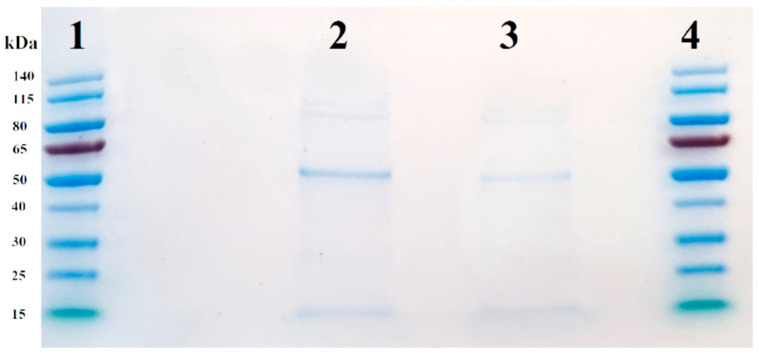
SDS-PAGE of control EVs and EV-AuNPs. Lanes 1 and 4: PageRuler Prestained Protein Ladder, lane 2: EV-AuNPs, and lane 3: empty control EVs.

**Figure 6 pharmaceutics-17-01015-f006:**
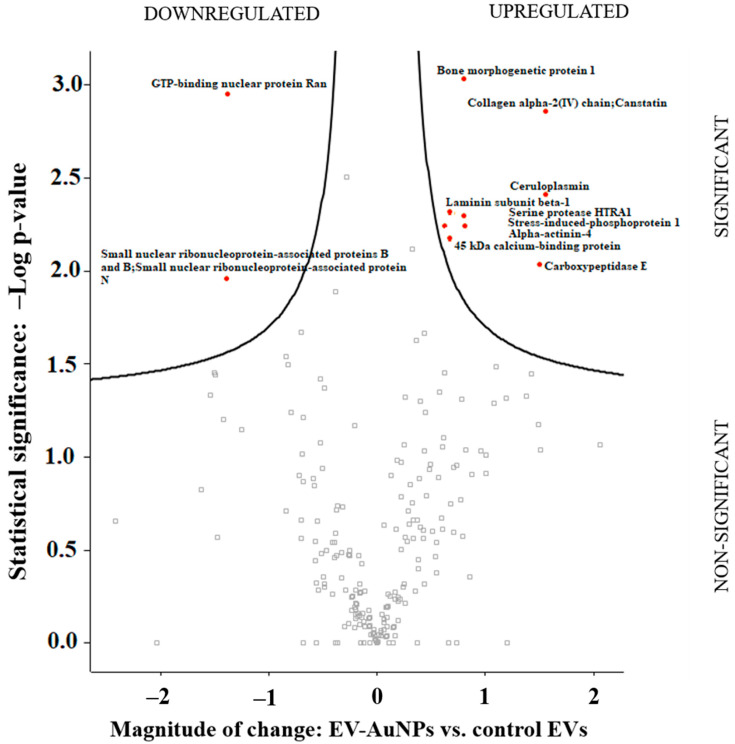
The Volcano plot obtained from the comparison of protein content between the EV-AuNPs and control EVs groups.

**Figure 7 pharmaceutics-17-01015-f007:**
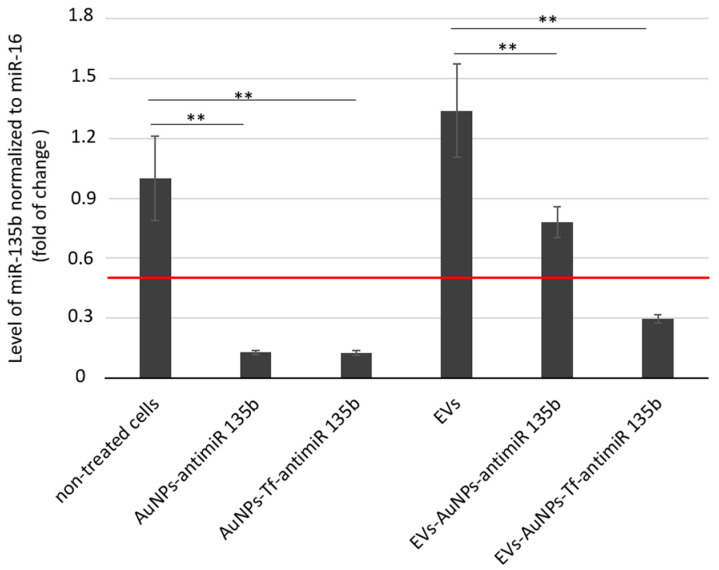
The inhibitory effect of antimiR-135b delivered into 4T1 tumor cells directly by AuNPs or via EVs-entrapped AuNPs. Tf—decorated AuNPs were tested in parallel. The effect of AuNPs carrying antimiR-135b and with/without Tf was evaluated with respect to non-treated cells. The effect of AuNPs-antimiR-135b with/without Tf loaded within EVs was evaluated with respect to cells that were incubated with control, empty EVs (sample marked as EVs). Level of target miR-135b was normalized using miR-16 internal control and the fold of change was assessed by standard 2^−ddCt^ algorithm. The red line marks fold of change = 0.5. ** marks significance of *p* < 0.01.

**Figure 8 pharmaceutics-17-01015-f008:**
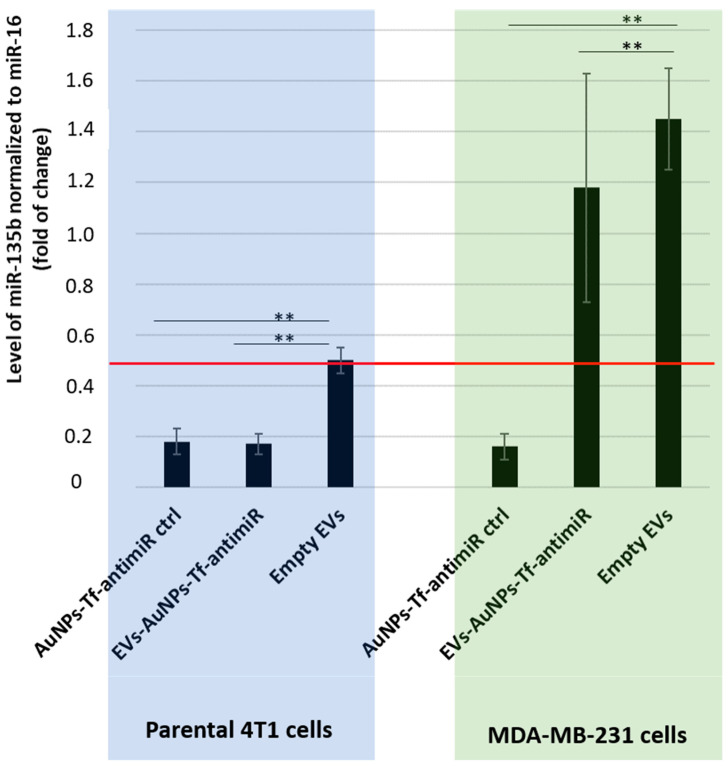
The inhibitory effect of the AuNPs-Tf-antimiR-135b on the level of miR-135b in the co- culture system before and after internalization in the 4T1 derived EVs. Level of target miR-135b was normalized using miR-16 internal control and the fold of change was assessed by standard 2^−ddCt^ algorithm. The red line marks fold of change = 0.5. All the possible significance differences between groups are marked as ** (*p* < 0.01).

**Figure 9 pharmaceutics-17-01015-f009:**
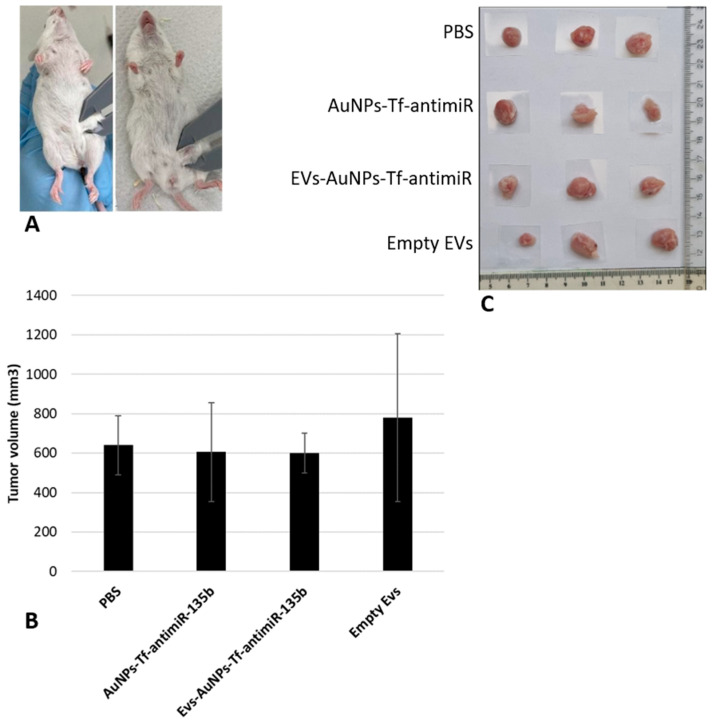
The analysis of the tumor diameters for each group includes (**A**) initial analysis of the tumors before the injection, (**B**) obtained tumor volume in different groups, and (**C**) the physical shape of the tumors. The extracted tumors from three mice in each group are shown as examples here.

**Figure 10 pharmaceutics-17-01015-f010:**
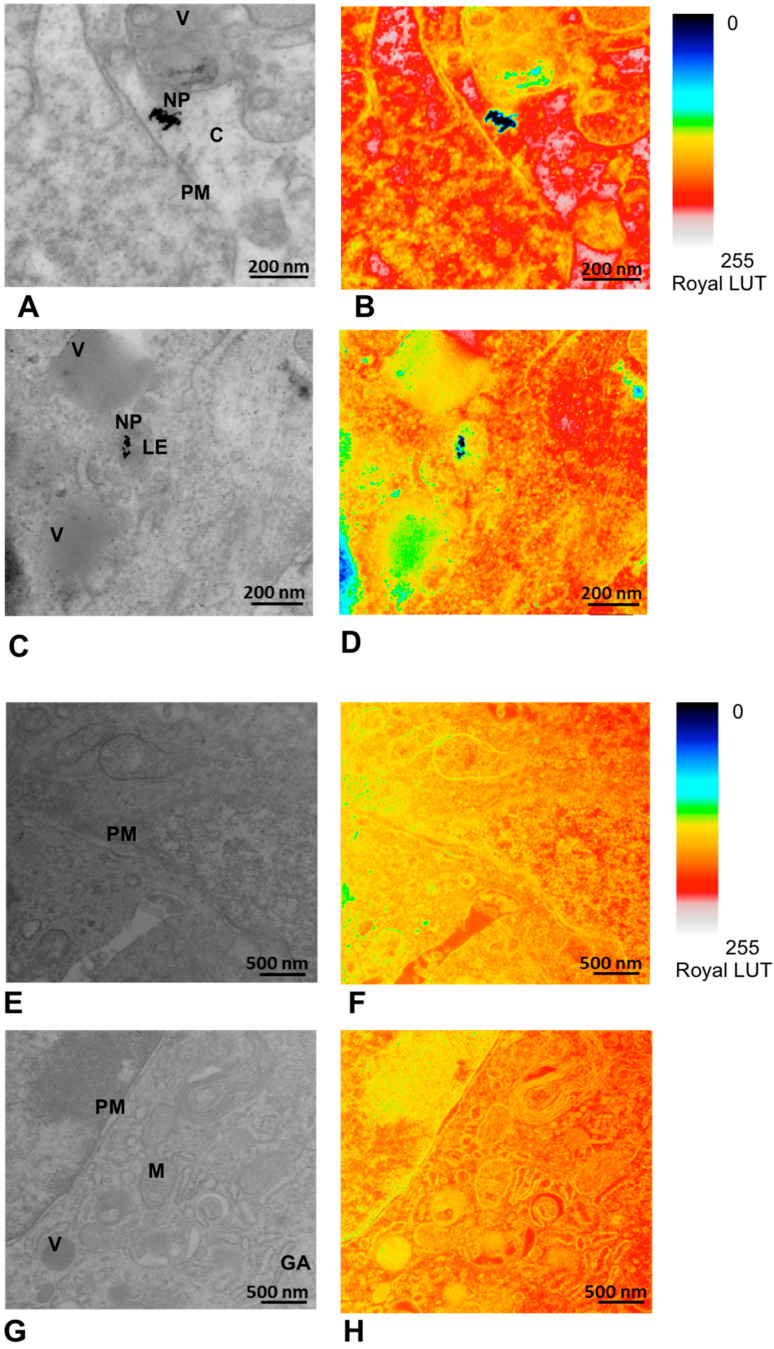
Detection of the AuNPs within the tumor tissue after in vivo application and its pseudocolor mapping in Royal LUT with the NP signal in black. (**A**,**B**) Tissue sample after administration of AuNPs-Tf-antimiR (positive control). The scale bar is 200 nm. The AuNPs are located in cell cytoplasm. (**C**,**D**) Tissue sample after administration of EVs containing AuNPs-Tf-antimiR (test group). The scale bar is 200 nm. The AuNPs are seen inside the late endosome. (**E**,**F**) Tissue sample from the animal that received PBS (negative control). The scale bar is 500 nm. (**G**,**H**) Tissue sample from animal that received empty EVs (negative control). The scale bar is 500 nm. PM—plasma membrane, C—cytoplasm, LE—late endosome, NP—AuNPs, V—vesicle, M – mitochondria, GA—Golgi apparatus.

**Figure 11 pharmaceutics-17-01015-f011:**
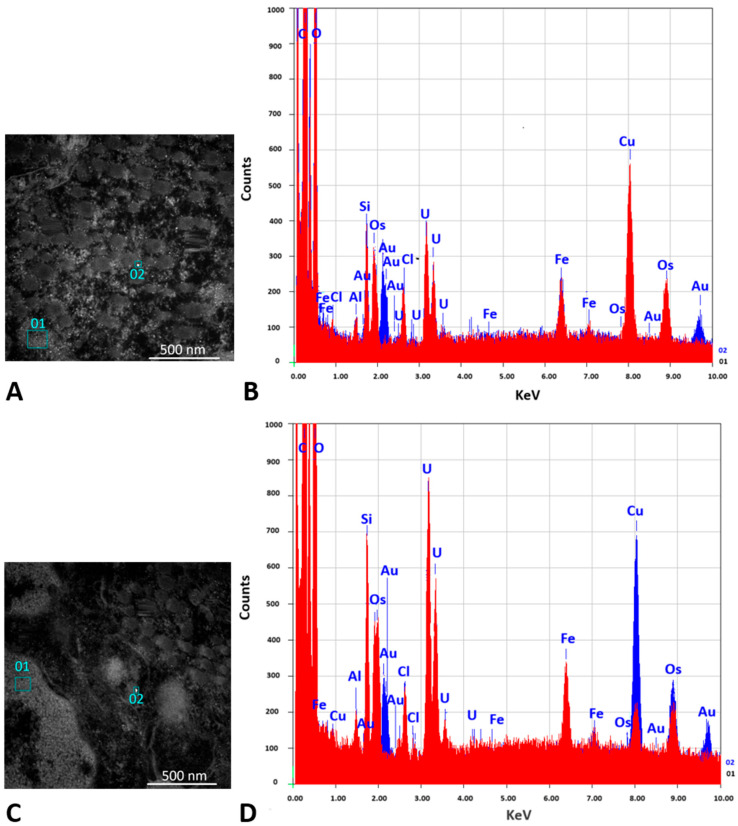
Confirmation of the AuNPs presence in the TEM image by EDS analysis. (**A**,**B**) Tissue sample after administration of AuNPs-Tf-antimiR (positive control). (**C**,**D**) Tissue sample after administration of EVs containing AuNPs-Tf-antimiR (test group). (**A**,**C**) show two selected areas of the grid that were used for EDS analysis; squares 01 were chosen from potential background and squares 02 were chosen from areas with potential AuNPs signals. Scale bar = 500 nm. (**B**,**D**) show the EDS spectrum obtained from squares 01 (red) and 02 (blue). The presence of elemental gold is observed in squares number 02.

**Figure 12 pharmaceutics-17-01015-f012:**
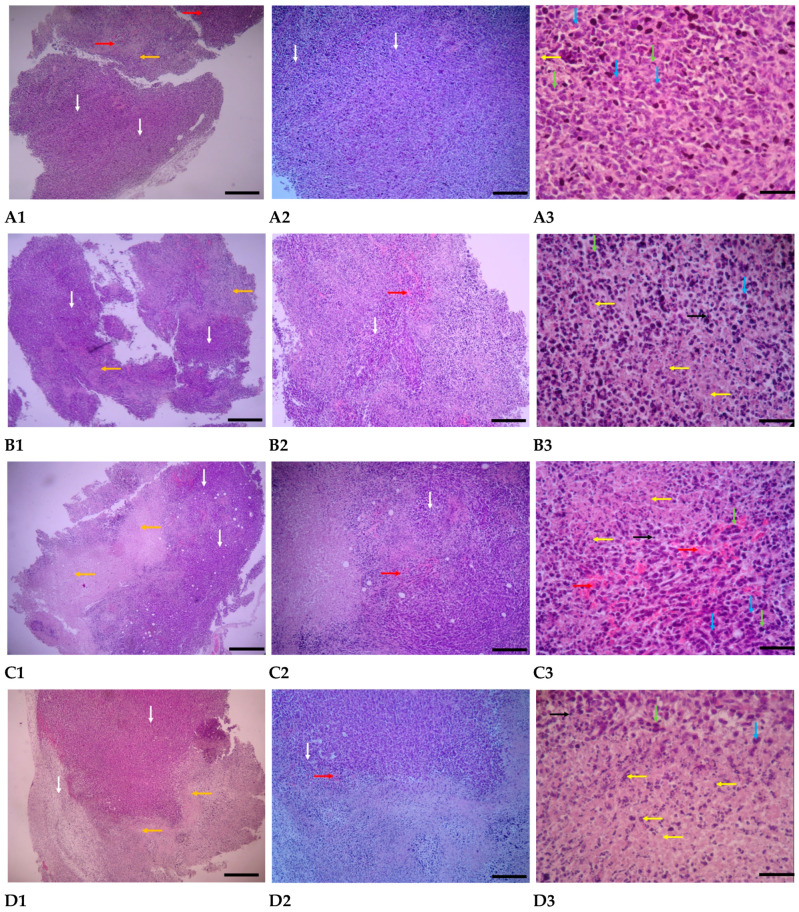
Tumor evaluations were conducted under a light microscope after staining using the H & E method. (**A1**–**A3**) represents the group that was treated with PBS; (**B1**–**B3**) represents the group that was treated with AuNPs-Tf-antimiR-135b; (**C1**–**C3**) represents the group that was treated with empty EVs; and (**D1**–**D3**) represents the group that was treated with EVs-AuNPs-Tf-antimiR-135b. The scale bars are 100 µm, 25 µm, and 10 µm in panels 1, 2, and 3, respectively. White arrows indicate tumor areas, blue arrows indicate neoplastic cells, green arrows indicate mitotic cells, orange arrows indicate necrotic areas, yellow arrows indicate necrotic and apoptotic cells, black arrows indicate inflammatory cells, and red arrows indicate hyperemia.

**Figure 13 pharmaceutics-17-01015-f013:**
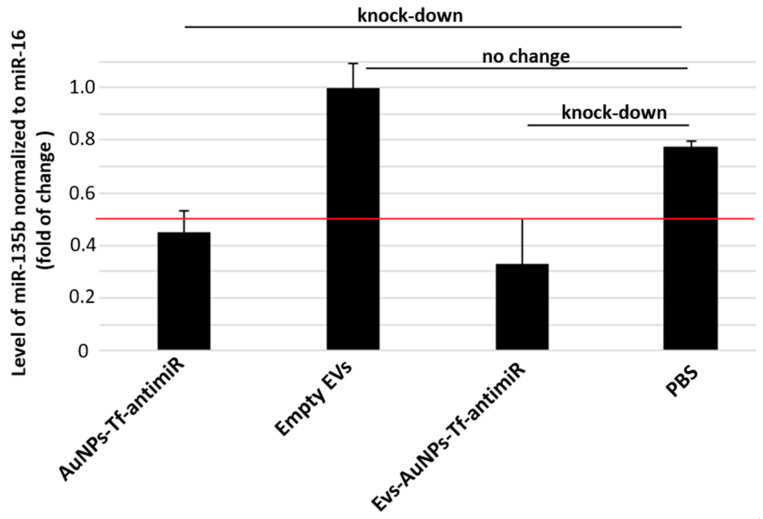
The inhibitory effect of the EV-containing AuNPs-Tf-antimiR-135b on the level of miR-135b in the experimental mouse model. Level of target miR-135b was normalized using miR-16 internal control and the fold of change was assessed by standard 2^−ddCt^ algorithm. The red line marks fold of change = 0.5 and remarkable knock-down of target antimiR is marked.

**Figure 14 pharmaceutics-17-01015-f014:**
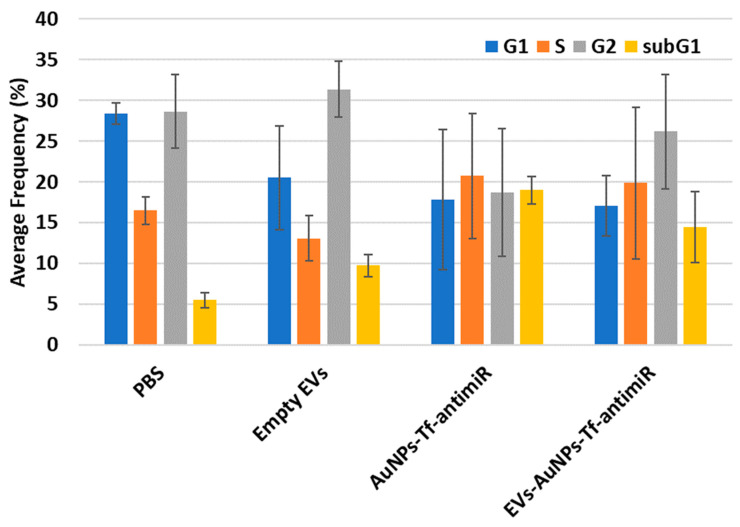
The cell cycle analysis results for the four tested groups. The sub-G1 phase which corresponds to the apoptotic cells is shown in the yellow column.

**Table 1 pharmaceutics-17-01015-t001:** Total amounts of AuNPs determined by GF-AAS.

Name of the Sample	Set 1 (µg)	Set 2 (µg)	Set 3 (µg)	Mean ± SD
AuNPs initial amount	58.70	58.70	58.70	
AuNPs amounts remained outside the cells (after 5 h incubation)	5.23	6.82	6.02	**6.02 ± 0.65**
AuNPs internalized by cells	53.47	51.88	52.68	**52.68 ± 0.65**
AuNPs amounts remained inside the cells (after overnight incubation)	15.20	13.20	13.70	**14.03 ± 0.85**
Exocytosed AuNPs inside the EVs	38.27	38.68	38.98	**38.64 ± 0.29**

**Table 2 pharmaceutics-17-01015-t002:** Size of the EVs evaluated by Zetasizer. Two peaks with different intensities were detected. Control EVs stands for EVs without AuNPs and the EV-AuNP stands for EVs loaded with AuNPs.

	Z-Average (nm)	PI	Peak 1 (nm)	Peak 1 Intensity (%)	Peak 2 (nm)	Peak 2 Intensity (%)
**Control EVs**	188.75 ± 1.5	0.37	262.7 ± 14	86.5 ± 0.3	41.48 ± 0.3	9.75 ± 0.8
**EV-AuNPs**	209.05 ± 19	0.27	268.2 ± 76	91.9 ± 1.4	58.96 ± 7.7	6.31 ± 1.0

**Table 3 pharmaceutics-17-01015-t003:** The list of the proteins depicted in the Volcano plot that exhibited significant differences between EV-AuNPs and control EVs. Protein ID corresponds to an identifier in the UniProt database (www.uniprot.org (accessed on 31 July 2025)).

Majority Protein IDs	Protein Names
**Upregulated in EV-AuNPs**	
O43707	Alpha-actinin-4
P00450	Ceruloplasmin
P07942	Laminin subunit beta-1
P08572	Collagen alpha-2(IV) chain; Canstatin
P13497	Bone morphogenetic protein 1
Q92743	Serine protease HTRA1
P16870	Carboxypeptidase E
Q9BRK5	45 kDa calcium-binding protein
P31948	Stress-induced-phosphoprotein 1
**Downregulated in EV-AuNPs**	
P14678	Small nuclear ribonucleoprotein-associated proteins B
P63162	Small nuclear ribonucleoprotein-associated protein N
P62826	GTP-binding nuclear protein Ran

**Table 4 pharmaceutics-17-01015-t004:** The output of LC-MS analysis of EVs containing AuNPs-Tf-antimiR 135b to indicate the presence of Tf in the EVs. The obtained Tf is from Homo sapiens, which is not present in the mice model (i.e., 4T1 cells derived from the mouse BALB/c strain).

Protein Group	Protein ID	Accession	Coverage (%)	Avg. Mass	Description
34	10133	P02787|TRFE_HUMAN	11	77064	Serotransferrin OS = Homo sapiens OX = 9606 GN = TF PE = 1 SV = 3

**Table 5 pharmaceutics-17-01015-t005:** Separation and grading of changes in variables evaluated in breast tumors. The observed changes are graded from 0 to 3. Grade 0 indicates no change, grade 1 indicates a slight increase, grade 2 indicates a moderate increase, and grade 3 indicates a severe increase.

Increased Inflammatory Cells	Increased Hyperemia	Expansion of Necrotic and Apoptotic Areas	Expansion of Neoplastic Areas	Groups
0	2	1	3	PBS
1	2	2	2	AuNPs-Tf- antimiR
1	2	1	2	Empty EVs
1	1	2	2	EVs-AuNPs-Tf antimiR

## Data Availability

The raw data supporting the conclusions of this article will be made available by the authors on request.

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
