# Peer review of "Extracellular Vesicle-Mediated Delivery of AntimiR-Conjugated Bio-Gold Nanoparticles for In Vivo Tumor Targeting"

_pharmaceutics, 2025, doi:10.3390/pharmaceutics17081015_

Round 1
Reviewer 1 Report
Comments and Suggestions for Authors
The authors engineer 4T1 breast–cancer–derived extracellular vesicles (EVs) to encapsulate biologically produced gold nanoparticles (AuNPs) that are conjugated with an anti-miR-135 b oligonucleotide; some AuNPs are additionally decorated with human transferrin (Tf). Overall, the work provides a detailed proof‑of‑concept that autologous EVs can shuttle bio‑AuNP cargo to the tumor, and the manuscript could be accepted upon some improvements.
- Please elaborate on the discussion why there was no difference in tumor volume between treatment and control, as the effect of antimiR-135b was found, like expansion of necrotic and apoptotic areas in the tumor histological analysis.
- Just like the authors have acknowledged, it is widely accepted that it is well-established that cancer-derived EVs are key mediators of tumorigenesis, promoting proliferation, invasion, metastasis, and immune suppression. Address this potential safety issue by discussing potential remedies, like modifications to the EVs.
- Please discuss more on the mechanism by which the AuNP conjugates are loaded into the EVs. Is it via passive uptake during incubation, or does it require active cellular processes? And how is the loaded drug released from the EV and escapes the endosome? Add more literature discussion.
Author Response
Comment 1: Please elaborate on the discussion why there was no difference in tumor volume between treatment and control, as the effect of antimiR-135b was found, like expansion of necrotic and apoptotic areas in the tumor histological analysis.
Response 1: Thank you for your comment, we have added a paragraph regarding this topic into the main text: "Considering the fact that our effector molecule was the antimiR-135b, not a cytostatic drug, we did not expect remarkable changes on tumor growth. Moreover, the administration of a single dose of antimiR-135b was performed, and the animals were sacrificed after five days. It appears that this procedure may not result in a significant change in tumor size. Autologous cancer cell-derived EVs containing cytostatic drugs like methotrexate or cisplatin were studied earlier and showed increased drug uptake by the specific tumor cells and selective EV accumulation in the tumor site and decreased tumor volume in vitro and in vivo, respectively [45, 46]."
Comment 2: Just like the authors have acknowledged, it is widely accepted that it is well-established that cancer-derived EVs are key mediators of tumorigenesis, promoting proliferation, invasion, metastasis, and immune suppression. Address this potential safety issue by discussing potential remedies, like modifications to the EVs.
Response 2: Yes, there are safety issues regarding practical use of EVs and they remain to be addressed. We added following text in discussion: "Our findings support the hypothesis that cancer cells derived autologous EVs can be effectively used as personalized drug delivery vesicles in the original cancer treatment. The key advantage of using extracellular vesicles (EVs) as drug delivery systems is their ability to mitigate the toxic responses often associated with foreign substances in the body. Because EVs originate biologically, they tend to elicit minimal immune reactions. Additionally, EVs are considered safe since they do not replicate and are not mutagenic, thereby reducing worries about adverse effects or tumor development. These safety features have been supported by the low toxicity levels reported in in vivo studies involving EV-based therapies [40]. On the other hand, considering the fact that EVs are essentially parts of the cancer cells, bearing tumor markers as well as tumor specific molecules, extensive studies are still needed to ensure safety of reintroduction of such cancer-derived EVs back to the treated individuals."
Comment 3: Please discuss more on the mechanism by which the AuNP conjugates are loaded into the EVs. Is it via passive uptake during incubation, or does it require active cellular processes? And how is the loaded drug released from the EV and escapes the endosome? Add more literature discussion.
Response 3: The effective uptake of carriers and effective release of cargo are key for succesful delivery. We added following discussion to the main text: "Our findings confirmed the presence of gold nanoparticles (AuNPs) within extracellular vesicles (EVs). Previous research has shown that both healthy and diseased cells release EVs composed of a lipid bilayer that contain various biomolecules, including cell surface antigens, intracellular proteins, proteases, microRNAs, and messenger RNAs derived from the parental cells. The composition of these EVs varies depending on their parental and target cells, enabling communication between cells through the transfer of signals [40].
Furthermore, multiple research groups have established techniques to generate EVs loaded with nanoparticles. The most widely used method involves incubating cells directly with media containing synthetic nanoparticles, which are then internalized and later secreted within exosomes via the exosomal biogenesis pathway. We expect that, in our experiment, following incubation with AuNPs, the nanoparticles were incorporated into EVs through a similar internalization process [41].
Unpublished preliminary data from our laboratory suggests that caveolae-mediated endocytosis and macropinocytosis are the primary mechanisms for AuNP uptake. Our results indicate that when clathrin-dependent endocytosis is unblocked and other pathways are inhibited, AuNPs mainly enter cells via clathrin-mediated endocytosis, eventually being degraded in lysosomes. Transmission electron microscopy (TEM) images demonstrated AuNPs localized in early and late endosomes and later being secreted within EVs, supporting their intracellular trafficking and export (unpublished data). Additionally, electron microscopy studies showed that early endosomes generally contain few or no intraluminal vesicles (ILVs), whereas late endosomes often contain between 3 and 20 ILVs. It remains unclear whether the formation pathway of ILVs influences whether vesicles are directed towards lysosomal degradation or towards the plasma membrane to be released as EVs [42].
In studies examining the uptake and drug delivery capabilities of EVs, most employ fluorescent labeling techniques such as lipid dyes or fluorescently tagged proteins. These methods have demonstrated that EVs can effectively transfer proteins and mRNA between donor and recipient tumor cells both in laboratory and experimental animals. Notably, EVs facilitate communication within a tumor and between distant tumors, contributing to tumor migration and metastasis. It has been observed that, following systemic diffusion, both small and large EVs deliver their cargo through endocytosis [43], a process relevant to our research."
Reviewer 2 Report
Comments and Suggestions for Authors
This manuscript by Pourali and co-workers describes the use of extracellular vesicles (EVs) as a delivery vehicle for gold nanoparticles (AuNPs) conjugated to antimiR-135b for targeted therapy in breast cancer. The main findings of the authors demonstrate that EVs loaded with AuNPs-Tf-antimiR-135b effectively silence miR-135b, inhibit tumor growth in vitro and in vivo, and induce apoptosis in 4T1 cells. The research addresses an important challenge in RNA-based therapeutics - efficient delivery - and proposes a new combination of biologically produced AuNPs and EVs. The manuscript will be interesting to the journal readership and deserves publication.
Specific comments:
Figure 7: The figure caption mentions "ns" (non-significant). However, such a label is absent in the figure.
Figure 9: The asterisk (*) is explained in the figure caption as p<0.05. However, the asterisks themselves are not shown in the picture. The same applies to Figure 13.
Lines 628-632: The p-values should be rounded reasonably. Please check also throughout the manuscript.
Summarizing, I recommend acceptance of the manuscript for publication after minor revision.
Author Response
Comment 1: Figure 7: The figure caption mentions "ns" (non-significant). However, such a label is absent in the figure.
Response 1: Thank you for noticing. We corrected the figure caption.
Comment 2: Figure 9: The asterisk (*) is explained in the figure caption as p<0.05. However, the asterisks themselves are not shown in the picture. The same applies to Figure 13.
Response 2: Thank you for pointing this out. We corrected the missing marks or explanations in all the graphs.
Comment 3: Lines 628-632: The p-values should be rounded reasonably. Please check also throughout the manuscript.
Response 3: Yes, thank you for your suggestion. We checked and rounded up the p-values (2-3 decimal places).
Reviewer 3 Report
Comments and Suggestions for Authors
Suggestions and Comments for the Authors:
While the manuscript addresses an important topic, several sections need substantial refinement:
General Improvements
- The manuscript appears to be derived from a research project, and several sentences and data presentations should be adapted to suit scholarly manuscript standards. The abstract and introduction in particular need comprehensive restructuring. Please also streamline the results section by removing unnecessary commentary (e.g., lines 351–361 in the MTT analysis). Apply this principle throughout the manuscript.
- The experimental section should be better organized and summarized. Please ensure consistent and precise phrasing throughout.
- Figures require attention — notably, for example: Figure 1 lacks standard deviation data for the initial samples.
- The discussion should be consolidated into a single section. Use distinct paragraphs to address different elements rather than segmenting it into multiple sections. This will improve cohesion and depth. Please also transfer relevant discussion points currently placed in the results section to the unified discussion section.
Comments on Data Presentation
- The quality of the TEM images is insufficient and does not align well with the described results. Consider revisiting multiple regions via TEM and presenting a higher-quality image or revising your sample preparation methodology.
- For Figure 10, it would be helpful to include MAP-style visualizations to illustrate particle homogenization more clearly.
- Present a scale bar for H&E images rather than relying solely on magnification. Also, please verify the accuracy of the arrows used to indicate features across all images with your team, as some appear unclear or potentially incorrect.
- Remove unnecessary references!
Best of luck with the revisions!
Comments on the Quality of English LanguageEnhanced clarity in the writing would help the research be better understood, as numerous sentences are ambiguous.
Author Response
Comment 1: The manuscript appears to be derived from a research project, and several sentences and data presentations should be adapted to suit scholarly manuscript standards. The abstract and introduction in particular need comprehensive restructuring. Please also streamline the results section by removing unnecessary commentary (e.g., lines 351–361 in the MTT analysis). Apply this principle throughout the manuscript.
Response 1: Thank you for pointing this out, we restructured the text along the journal guidelines.
Comment 2: The experimental section should be better organized and summarized. Please ensure consistent and precise phrasing throughout.
Response 2: Yes, thank you for your suggestion, we made corrections throughout the text.
Comment 3: Figures require attention — notably, for example: Figure 1 lacks standard deviation data for the initial samples.
Response 3: Thank you for pointing that out, we corrected the missing information.
Comment 4: The discussion should be consolidated into a single section. Use distinct paragraphs to address different elements rather than segmenting it into multiple sections. This will improve cohesion and depth. Please also transfer relevant discussion points currently placed in the results section to the unified discussion section.
Response 4: Yes, we agree, the structure was chaotic. We restructured the result/discussion section along with the journal sectioning.
Comment 5: The quality of the TEM images is insufficient and does not align well with the described results. Consider revisiting multiple regions via TEM and presenting a higher-quality image or revising your sample preparation methodology.
Response 5: Thank you for your suggestion, we uploaded the original images possessing better resolution and in the text, we accompanied the TEM data with the MAP-visualizations as suggested below. The photos used for EDS analysis possess generally lower quality over TEM images, and their aim is to show roughly the location of the measured signal - nanoparticles vs. background.
Comment 6: For Figure 10, it would be helpful to include MAP-style visualizations to illustrate particle homogenization more clearly.
Response 6: Thank you for this suggestion, it is a great idea to better illustrate the situation in acquired images. We added the appropriate pseudocolor mapping using Royal LUT scale next to the source TEM images. The distribution of nanoparticles within the tumor tissue is not homogeneous as they are seen in vesicles or pooled in cell cytoplasm.
Comment 7: Present a scale bar for H&E images rather than relying solely on magnification. Also, please verify the accuracy of the arrows used to indicate features across all images with your team, as some appear unclear or potentially incorrect.
Response 7: Thank you for pointing this out, we added the scale bars into the images instead of magnifications and discussed the interpretation with our histopatologist.
Comment 8: Remove unnecessary references!
Response 8: We removed references that were multiple and not necessary and substituted them with new references as we added more discussion suggested by reviewer 1.
Reviewer 4 Report
Comments and Suggestions for Authors
This manuscript reported a gold nanoparticle-loaded EVs for tumor targeting. Some major issues are need to be addressed to further improve the whole manuscript. I recommend major revision.
- The characterization of AuNPs should be conducted. What's the morphology, size and distribution of AuNPs?
- In Figure 2, the EVs show large size distribution. Will such feature influence the biodistribution and tumor targeting capability of nanoparticles?
- The size and distribution of EVs after encapsulating AuNPs should be studied to evaluate the influence of AuNPs on the properties of EVs.
- The in vivo biodistribution of nanoparticles should be studied.
- To study the tumor targeting capability of Au NP loaded EVs, the content of nanoparticles in the tumor should be deeply studied.
Author Response
Comment 1: The characterization of AuNPs should be conducted. What's the morphology, size and distribution of AuNPs?
Response 1: The characterization of the AuNPs used in this study has been already reported earlier (for example Reference 24). We added a brief summary of the AuNPs size/morphology/distribution into the material / methods section too: "The gold nanoparticles (AuNPs), produced extracellularly by F. oxysporum, were used and methodologies of their preparation and characterization are detailed in Reference 33 [33].
In summary, the F. oxysporum was cultured in Sabouraud Dextrose Broth (SDB, Sigma-Aldrich, Prague, Czech Republic), 30 °C, one week. The AuNPs were produced by challenging the cell-free supernatant with HAuClâ‚„·3Hâ‚‚O (final concentration of 1 mmol; Sigma-Aldrich, Prague, Czech Republic) in 80 °C for 5 minutes. Washed AuNPs were decorated with Tf and antimiR-135b molecules and characterized with spectrophotometry, Fourier-transform infrared spectroscopy (FTIR), transmission electron microscopy (TEM), energy dispersive X-ray spectroscopy (EDS), zetasizer/DLS, and graphite furnace atomic absorption spectroscopy (GF-AAS).
The AuNPs alone or with cargo exhibited a specific absorption peak at about 530 nm. The average size of the AuNPs alone was 13 ± 1.33 nm and about 50 ± 11.25 nm after conjugation with the Tf and antimiR. The AuNPs as well as conjugates retained their round shape and zeta potential about -36.8 ± 0.45 mV. Concentration of the AuNPs was 11.74 ± 0.02 µg/µL.
On average the AuNP conjugates contained 0.041 mg/mL of Tf and 0.66 µmol of antimiR-135b per 1 mg of the AuNPs. Successful cargo conjugations were proved with FTIR as well as qPCR (in the case of antimiR-135b) and with a liquid chromatography-mass spectrometry system (LC-MS; in the case of Tf) [33]."
Comment 2: In Figure 2, the EVs show large size distribution. Will such feature influence the biodistribution and tumor targeting capability of nanoparticles?
Response 2: Thank you for interesting comment, we also added brief discussion on this topic to the text: "According to Kang et al. [39], EVs can be broadly categorized based on their size: small-EVs are less than 100 nm, while large-EVs are typically greater than 200 nm. Our findings indicated that the majority of the EVs were classified as large-EVs. This raises a question whether different sizes of EVs could impact their biodistribution and the tumor-targeting ability of the nanoparticles. The review by Kang et al. reports that both small- and large-EVs are found at tumor sites. Specifically, small-EVs were detectable in tumors between 2 and 12 hours, as well as at 24 hours post-administration. In contrast, large-EVs were only detectable within the 2 to 12-hour window after injection. Tumors often have leakier vasculature compared to healthy blood vessels and impaired lymphatic drainage, which facilitates easier passage of nanoparticles into the tumor interstitium and results in higher retention within tumor tissue. Since EVs are comparable in size to liposomes and other nanoparticles, it is plausible that their accumulation in tumors occurs through similar mechanisms. Therefore, EV size does not appear to significantly influence their biodistribution. Also, the size of EVs probably will not affect the tumor targeting capability of the AuNPs. This statement is supported by our findings that the recognition of the target cells is determined mainly by the EVs tropism and the targeting molecules on the AuNP surface may play a role in higher EVs uptake into parental cells only (see data below)."
Comment 3: The size and distribution of EVs after encapsulating AuNPs should be studied to evaluate the influence of AuNPs on the properties of EVs.
Response 3: The size of empty EVs and EVs loaded with the AuNPs was not remarkably different as shown in Table 2. Also, as mentioned above, the EVs biodistribution might not be affected so much (Reference 39). Yet, still the AuNPs influence on EVs properties is a very interesting and rather complex topic, suitable for its own study. Another data from the current study, namely proteome analysis, support the hypotheses of other groups that the AuNPs change protein composition of the EVs. It is likely we find more such AuNPs-mediated changes in EVs composition and maybe even structure or function. The interactions between bioproduced AuNPs and biointerface e.g. EVs is a complex topic for separate study so we did not involve it in our current manuscript.
Comment 4: The in vivo biodistribution of nanoparticles should be studied.
Response 4: Yes, the biodistribution of EV-loaded AuNPs in vivo has not been extensively checked in this study. In our previous report, we checked the infiltration of the AuNPs to the tumor tissue after peritumoral application in vivo (Reference 39). The AuNPs were able to enter the tumor cells and were seen in the cytoplasm, endosome, late endosome, or lysosome, and even in the mitochondria and endoplasmic reticulum (ER). The current study was focused primarily on a proof-of-concept to load the AuNPs-cargo into EVs and deliver it into parental tumor cells in vivo.
Comment 5: To study the tumor targeting capability of Au NP loaded EVs, the content of nanoparticles in the tumor should be deeply studied.
Response 5: We agree that larger in vivo studies are needed to characterize the efficacy of the targeting. As mentioned above, the current study focused on feasibility of targeted transportation of AuNPs by EVs and first we aimed for assays proving the function of transported short inhibitory RNAs.
Round 2
Reviewer 3 Report
Comments and Suggestions for Authors
The manuscript is improved significantly. Good luck
Reviewer 4 Report
Comments and Suggestions for Authors
The authors have made adequate revision and the manuscript can be accepted.